# Prompt2Rec : Prompt based user and item Re-characterizing method for Recommendation

## Abstract

Collaborative Filtering, which utilizes user-item interaction data is widely adopted in Recommendation Systems ; however, the lack of interaction data can adversely affect recommendation performance. To address this issue, research incorporating Natural Language Processing (NLP) has made progress in leveraging review texts that contain rich information about user preferences and item attributes. Nevertheless, the conventional approach of integrating the entire review text and using it as an input, which has been widely used in previous research, can be vulnerable to noise (i.e., data with little relevance to user preferences or item attributes). In this study, we propose a novel user and item re-characterizing method called Prompt2Rec, which introduces the Prompt-based learning paradigm of NLP. It generates key factors that newly defined essential user and item characteristics from review texts and uses them as new information to train the recommendation model. Through experiments, we demonstrate that our proposed method can generate intuitive key factors related to user preferences and item attributes from reviews, and we validate that using these key factors in model training leads to improved performance compared to existing methods that rely on review texts. Furthermore, we explore the potential of visualizing the model's attention weights on the key factors for providing explanations of recommendations.

## 1 Introduction

Collaborative Filtering (CF), utilizing user-item interaction data, such as ratings and clicks, is the most widely used in Recommendation System (RS). Despite its successful adoption, CF faces challenges when handling limited interaction data (cold-start problem) or data sparsity, leading to reduced prediction reliability (Su & Khoshgoftaar, 2009). To address these issues, researchers have attempted to utilize review texts as a new data source. Text reviews contain rich information regarding user preferences and item attributes that cannot be fully understood from ratings alone (Zheng et al., 2017). Recent advancements in NLP (Otter et al., 2021) have introduced them into RS. Several models have been proposed to effectively extract user and item latent features from review texts and have shown promising results (Wu et al., 2022).

Although these Review-based RS have shown promising performance, the conventional approach of integrating the entire review text written by users (or items) and using it as an input, which has been widely used in previous research, can be vulnerable to noise problems. In other words, (Even if word-level text preprocessing is performed,) if all review texts are used, there is likely to be a significant amount of irrelevant or unnecessary data (noise) unrelated to user preferences or item attributes. Using such information may hinder the model from adequately reflecting user preferences and item attributes in the latent features, leading to a decrease in recommendation performance. Reducing irrelevant or unnecessary data and extracting essential information from review texts can be the critical aspects of Review-based RS. Some previous studies attempted to address this problem by utilizing attention mechanisms to selectively prioritize important review texts (Seo et al., 2017) or by modeling reviews individually and then integrating them (Chen et al., 2018). However, these methods inherently rely on the use of irrelevant text data, which limits the resolution of this problem.

In this study, we propose a novel user and item re-characterizing method called Prompt2Rec, which introduces the Prompt-based learning paradigm of NLP (Song et al., 2023) to address the noise problem in review data. Our proposed method utilizes a pre-trained language model and prompts to

generate key factors that re-characterize users and items from review texts, and uses these key factors to train the recommendation model. In detail, Prompt2Rec consists of two main processes. First, Prompt-based user & item key factor generation, prompts are designed to enable the generation of key factors that newly define user preferences and item attributes from review text. These prompts are then utilized with the language model, RoBERTa (Liu et al., 2019b) to generate key factors. Second, Recommendation model learning, the generated key factors are used as new data source to train a model for predicting user's ratings for items.

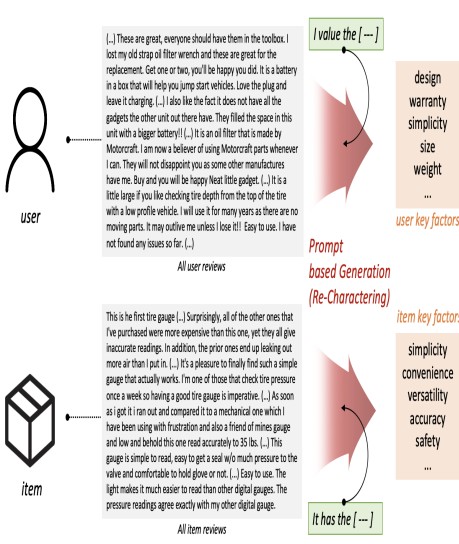

Figure 1: Illustration of key factor generation

As shown in Figure 1, the key factors generated by Prompt2Rec are essential words representing user preferences (or item attributes) that newly refine an integrated review text. By utilizing prompts with a language model pre-trained on large corpora, the generated key factors condense the essential information from the reviews, thus addressing the noise problem mentioned earlier. Moreover, they offer an intuitive understanding of user preferences and item attributes, which are difficult to grasp directly from review texts. Our method uses the key factors as new information to train the recommendation model, and our experimental results demonstrated that our method outperform the conventional approach that uses integrated review text. Furthermore, we explore the possibility of enhancing the explainability of the model's recommendations by applying attention to the generated key factors and visualizing attention weights. To the best of our knowledge, this study is the first Recommendation System that applies Prompt-based learning paradigm in NLP to generate user and item characteristics from reviews and uses them as new data sources.

## 2 RELATED WORK

### 2.1 REVIEW-BASED RECOMMENDATION SYSTEM

CF suffers from issues of data sparsity and cold-start problem, which can negatively affect recommendation performance. Researchers have explored the use of additional information relevant to the interaction data. Review data, which were easily obtained owing to the development of e-commerce and social platforms, were considered first because they included reasons for ratings, user preferences, and item attributes. Early studies used Topic Modeling to extract latent topics from reviews and integrated them with interaction data for modeling (Nsir et al., 2022). However, these methods lose a large amount of contextual semantic information from reviews (Zheng et al., 2017). In addition, most studies predicted ratings using linear methods, which are limited to modeling complex relationships among features (He et al., 2017).

With the advancement of NLP, related techniques have been introduced into review-based RS and have become state-of-the-art techniques. These studies mainly used convolutional neural networks (CNNs) to extract contextual semantic information from review texts (Kim, 2014). DeepCoNN (Zheng et al., 2017), which was first proposed as a Review-based deep learning model, uses CNNs to embed user and item latent features from the integrated review text of users and items. These latent features pass through a Factorization Machine (FM) to predict the ratings (Rendle, 2012). Since then, various models have been proposed to improve the embedding of user and item features from reviews to provide better recommendations. Representatively, several models utilize attention modules for feature extraction to obtain word importance and interpretability (Seo et al., 2017; Li et al., 2019; Liu et al., 2019a). Other models employ knowledge distillation by separating the network learning target reviews (Catherine & Cohen, 2017). some models leverage user and item rating patterns as additional information (Liu et al., 2020; Xi et al., 2022), and others consider aspects of user and item features (Chin et al., 2018).

Typically, these models integrate the entire review text written by users (or items) and use it as training input. However, this approach can be sensitive to noise problems in the review texts. The use of irrelevant or unnecessary information can cause the model to not properly reflect user preferences and item attributes in the latent features, leading to a decrease in recommendation performance.

## 2.2 PROMPT-BASED LEARNING

Prompt-based learning is a Zero-shot or Few-shot learning method proposed to maximize the utilization of pretrained language model knowledge in NLP (Song et al., 2023). Unlike Supervised learning, in which models are trained to take inputs and predict outputs, Prompt-based learning is based on language models that model word probabilities. In contrast to fine-tuning, which re-trains the language model using task-specific objective functions for downstream tasks and labeled data, Prompt-based learning transforms downstream tasks into language model tasks (i.e., word sequence prediction) by using prompts fed into the language model without the need for objective functions. To design prompts, templates generally consist of a sentence that induces the desired output and a blank token filled with the output. For example, given the sentence "The movie was interesting," to make a prompt that infers the sentiment of the sentence, the template "I felt [MASK]" is appended to the sentence, and then the prompt is fed into the language model. When the language model infers the [MASK], the inferred word token is matched with the target label. For example, if the word token "fun" is inferred, it matches the target label "positive" (Liu et al., 2023).

Starting with GPT-2 (Radford et al., 2019), Prompt-based learning has been actively researched, and subsequent studies have developed various forms of prompts by altering their generation methods and styles (Li & Liang, 2021; Shin et al., 2020). Several methods have been proposed to enhance task performance (Schick & Schütze, 2021a;b; Gao et al., 2021). In particular, using prompts, LAMA (Petroni et al., 2019) identified that pre-trained language models trained on vast corpora of data possess substantial knowledge and common sense. In contrast to the majority of previous studies, where the language model's inferred words are matched with target labels, our study uses the word itself; the words inferred by the language model through prompts are used as key factors to indicate user preferences and item attributes.

## 3 PROPOSED METHOD

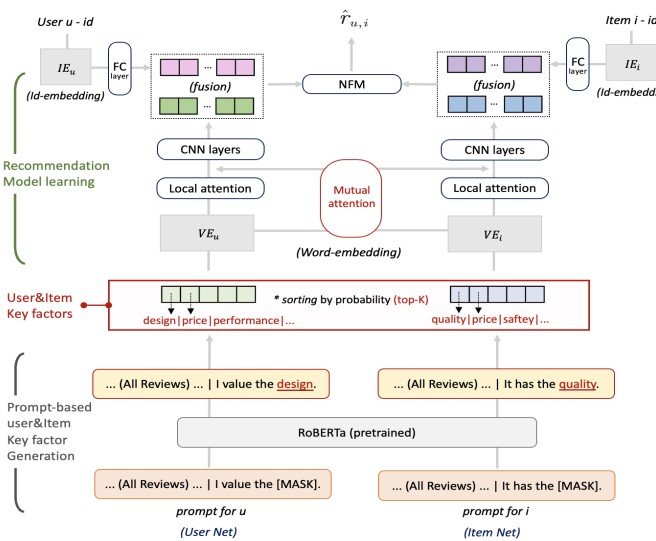

Figure 2: Overall process of the proposed method

Figure 2 shows the overall process of the proposed method, which can be divided into two parts. First, Prompt-based user & item key factor generation: In Prompt-based learning using a pretrained language model (RoBERTa) and prompts, we generate key factors that newly define user preferences and item attributes from integrated reviews. Second, Recommendation model learning: the generated key factors are used as new data sources to train a model for predicting the user's ratings of items. In our method, the user and item networks follow the same process, with differences only in the input information. Therefore, we focused on explaining the user modeling process.

### 3.1 PROMPT-BASED USER AND ITEM KEY FACTOR GENERATION

We utilize a pre-trained language model (RoBERTa) to generate the key factors. The user-integrated reviews $d^u$ denote a set of reviews written by user $u$ formed by merging the total number $n$ of words.

$$d_{1:n}^u = [\, d_1^u, \oplus d_2^u, \oplus \cdots \oplus \, d_n^u \,] \tag{1}$$

Given $d^u$, a template with a mask token is added to define the user preferences, forming a prompt, as shown in Figure 3. The prompt is designed by considering the tokenizer and special tokens of the language model being used, ensuring that it does not exceed the maximum number of input tokens of the language model (e.g., 512 tokens for RoBERTa).

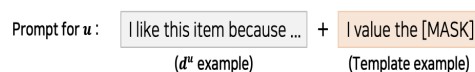

Figure 3: Prompt composition example

The prompt is input into RoBERTa, and then we obtain the inference probabilities $P([MASK] = v \,|Prompt_u)$ of the word token $v$ at the mask token position. The range of word token $v$ is equal to the token count used by the language model's tokenizer. The word token list is sorted in the descending order of inference probabilities, and the top $k$ word tokens with the highest inference probabilities (representing user preferences) are selected to form the key factor set $\hat{v}^u$.

$$v_{1:k}^u = [\, \hat{v}_1^u, \hat{v}_2^u, \, \ldots \, \hat{v}_k^u \,] \tag{2}$$

### 3.2 RECOMMENDATION MODEL LEARNING

The generated key factor set $\hat{v}^u$ is used to train the rating prediction model. Because user-item key factors define user preferences and item attributes, we expect higher ratings when there is a good match between user-item key factors. We assume that the semantic similarity between user and item key factors is important in predicting ratings, and we aim to model this in our method.

For example, if the user key factor set is [price, quality, design], a higher rating can be expected when the item key factor set is [price, performance, quality] compared to when it is [strength, simplicity, safety] because the user preference and item attributes are more similar in the former.

Our method uses a small number of refined key factors to learn the latent features. Experimentally, we observed that increasing the complexity of the model did not improve the performance. Accordingly, we used the minimum module to ensure the modeling concept and explainability. Inspired by existing research (Liu et al., 2019a; Seo et al., 2017), we propose a CNN architecture that utilizes Mutual Attention (to calculate weights considering the similarity between user and item key factors) and Local Attention modules (to assign higher weights to important key factors).

#### 3.2.1 WORD-EMBEDDING

Given a set $\hat{v}^u$ consisting of k key factors defining user $u$, each key factor in $\hat{v}^u$ is mapped into word vector $g^u$ using a pre-trained word embedding Glove (Pennington et al., 2014). User word vector matrix $G^u \in \mathbb{R}^{k \times d}$, where $d$ is the number of dimensions of the word embedding, is formed by combining the word vectors $g^u \in \mathbb{R}^d$

$$\begin{aligned} G^u &= [g_1^u, g_2^u, \cdots, g_k^u] \\ G^i &= [g_1^i, g_2^i, \cdots, g_k^i] \end{aligned} \tag{3}$$

#### 3.2.2 MUTUAL ATTENTION

We utilize the Mutual Attention module to consider the similarity between user-item key factors. The correlation scoring function measures the Euclidean distance between the user word vector $g^u$ and the item word vector $g^i$ to calculate the similarity (relevance) between the word vectors.

$$f_{correlation\ score} = 1/(1 + |g_p^u - g_q^i|) \tag{4}$$

Using the correlation scoring function, we calculate the user-item pair Mutual Attention matrix $A$. Each element of matrix $A$ represents the similarity (relevance) between each user word vector $g_p^u$ and item word vector $g_q^i$. Each row of matrix $A$ denotes the similarity (relevance) of each user word vector $g_p^u$ to the item word vector $g_q^i$ and vice versa for each column.

$$A = f_{correlation-score}(G^u, G^i) \tag{5}$$

We calculate the similarity weights $a^u(p)$ for the user word vector $g_p^u$ and $a^i(q)$ for the item word vector $g_q^i$ as follows. For example, if the weight $a^u(p)$ is relatively large, then the user key factor (word vector) is highly similar to the item key factor (word vector).

$$a^u(p) = softmax(\sum A[p, :])$$
$$a^i(q) = softmax(\sum A[:, q]) \tag{6}$$

### 3.2.3 Local attention

We utilize the Local Attention module to reflect the importance of individual key factors. We use Local Attention without applying a sliding window because there is no correlation between adjacent key factors. The importance weights $b^u(p)$ and $b^i(q)$ for each user word vector $g_p^u$ and item word vector $g_q^i$ are calculated as follows, where $w_{la}$ and $b_{la}$ denote the parameters and bias, respectively, and $*$ denotes the element-wise multiplication and summation operation.

$$b^u(p) = softmax(ReLU(g_p^u * w_{la}^u + b_{la}^u))$$
$$b^i(q) = softmax(ReLU(g_q^i * w_{la}^i + b_{la}^i)) \tag{7}$$

$b^u(p)$ denotes the importance of the p-th user key factor. If $b^u(p)$ is larger, then the model considers the p-th user key factor (word vector) to be more important than the other key factors (word vector). We then combine the calculated weights using mutual and local attention. We obtain the weighted user word vectors $\hat{g}_p^u$ by multiplying the similarity weight $a^u(p)$ by the importance weight $b^u(p)$ with the user word vector $g_p^u$. The weighted user word vector matrix $\hat{G}^u$ is formed by combining the weighted user word vectors $\hat{g}_p^u$ with their respective weights.

$$\hat{g}_p^u = a^u(p) \cdot b^u(p) \cdot g_p^u$$
$$\hat{g}_q^i = a^i(q) \cdot b^i(q) \cdot g_q^i \tag{8}$$

$$\hat{G}^u = [\,\hat{g}_1^u, \hat{g}_2^u, \cdots, \hat{g}_k^u\,]$$
$$\hat{G}^i = [\,\hat{g}_1^i, \hat{g}_2^i, \cdots, \hat{g}_k^i\,] \tag{9}$$

### 3.2.4 CNN Layers

After obtaining the weighted user word vector matrix $\hat{G}^u$, convolutional operations are used to extract contextual semantic information. The convolutional layer comprises $f$ convolutional filters, denoted by $K_j \in \mathbb{R}^{1 \times d}$ and is calculated using equation 10. Because individual user word vectors $\hat{g}^u$ based on user key factors are not related to neighboring user word vectors, we set the kernel size of the convolutional filters to 1, allowing effective extraction of individual features.

$$c^j = ReLU(\hat{G}^u * K_j^u + b_c^u) \tag{10}$$

After the convolution, the obtained feature maps, denoted by $c^j \in \mathbb{R}^k$, are passed through the max-pooling and fully connected layer to obtain the key factor-based user feature $\hat{h}^u \in \mathbb{R}^f$, where $W_{fc}$ and $b_{fc}$ denote the parameters and bias of the fully connected layer, respectively.

$$h^j = max(c_1^j, c_2^j, ..., c_k^j) \tag{11}$$

$$h^u = [h^1, h^2, ..., h^f] \tag{12}$$

$$\hat{h}^u = ReLU(W_{fc}^u \cdot h^u + b_{fc}^u) \tag{13}$$

### 3.2.5 FEATURE FUSION

We leverage user and item id information to capture better user preferences and item attributes. It is used as data for identifying users and items. As in previous research (He et al., 2017), id information is used to learn the latent features of each user and item, and then the id-based features are fused with the previously obtained key factor-based features. There are embedding matrices $\Omega_a$ and $\Omega_b$ for mapping user $u$ and item $i$ ids to n-dimensional representations. user id embedding $o^u$ goes through a fully connected layer as follows, where $\boldsymbol{W_{id\_fc}^u}$ and $b_{id\_fc}^u$ denote the parameters and bias.

$$\boldsymbol{o^u} = \boldsymbol{\Omega_a}(user_u)$$
$$\boldsymbol{o^i} = \boldsymbol{\Omega_b}(item_i)$$

(14)

$$\boldsymbol{\hat{o}^u} = ReLU(\boldsymbol{W_{id\_fc}^u} \cdot \boldsymbol{o^u} + b_{id\_fc}^u)$$

(15)

Finally, the user feature $\boldsymbol{u}$ and item feature $\boldsymbol{i}$ are obtained by adding the key factor-based feature $(\boldsymbol{\hat{h}^u}, \boldsymbol{\hat{h}^i})$ and id-based feature $(\boldsymbol{\hat{o}^u}, \boldsymbol{\hat{o}^i})$, and concatenating them to form the user-item feature $\boldsymbol{z}$.

$$\boldsymbol{u} = \boldsymbol{\hat{h}^u} + \boldsymbol{\hat{o}^u}$$
$$\boldsymbol{i} = \boldsymbol{\hat{h}^i} + \boldsymbol{\hat{o}^i}$$

(16)

$$\boldsymbol{z} = [\,\boldsymbol{u}, \boldsymbol{i}\,]$$

(17)

### 3.2.6 NEURAL FACTORIZATION MACHINE

We use Neural Factorization Machine (NFM) (He & Chua, 2017) to capture high-order nonlinear interactions. Rating prediction using the NFM is as follows:

$$\hat{r}_{u,i}(\boldsymbol{z}) = \ m_0 + \sum_{j=1}^{z} m_j z_j + \boldsymbol{h^T} ReLU(\boldsymbol{W_L}(...ReLU(\boldsymbol{W_1} f(\boldsymbol{z}) + b_1)...) + b_L)$$

(18)

$$f(\boldsymbol{z}) = \frac{1}{2}[(\sum_{j=1}^{|z|} z_j \boldsymbol{v_j})^2 - \sum_{j=1}^{|z|} (z_j \boldsymbol{v_j})^2]$$

(19)

$z_j$ is the j-th feature of the user-item feature $\boldsymbol{z}$, $m_0$, and $m_j$ denote the global bias and coefficient for $z_j$. $f(z)$ models the high-order interaction between features and $\boldsymbol{v_j}$ is the embedding vector for $z_j$ with $c$ dimensions. In the third term of equation 18, the nonlinear interaction is modeled by stacking fully connected layers on $f(z)$, where $\boldsymbol{h}$, $\boldsymbol{W_L}$, $\boldsymbol{W_1}$ are the parameters, and $b_L$, $b_1$ are the biases.

### 3.2.7 MODEL TRAINING

The objective function to train our recommendation model is as follows. The first term of the objective function minimizes the difference between the predicted rating $\hat{r}_{u,i}$ and actual rating $r_{u,i}$. The second term is the regularizer to prevent overfitting, where $\lambda$ is a regularization coefficient. The recommendation model learning part is trained end-to-end using a backpropagation technique.

$$J = \sum (\hat{r}_{u,i} - r_{u,i})^2 + \lambda ||\theta||^2$$

(20)

## 4 EXPERIMENTS

In this section, we evaluate the proposed method using publicly available Amazon review datasets. First, we describe the prompt design and results of the prompt-based user and item key factor generation, which is the first part of our method. Moving on to the second part of our method (recommendation model learning), we introduce the baseline method and experimental settings used for comparative evaluation and present the experimental results.

## 4.1 DATASETS

We used the Amazon product review 5-core dataset[1], which provides user reviews, ratings (McAuley et al., 2015) and selected five categories: Automotive, Amazon Instant Video, Office Products, Digital Music, Grocery and Gourmet Food. The data characteristics are listed in Appendix A.

The dataset was randomly divided into training, validation, and test sets. (80%, 10%, 10%) The data were pre-processed to ensure that each user and item had at least one rating. Review text preprocessing, such as punctuation removal and length adjustment (limiting the review length to 400 words), was performed. Unlike in other studies (Catherine & Cohen, 2017), we did not perform stop-word removal to input the entire sentence sequence into the language model.

## 4.2 DESIGNING PROMPT

Prompts can be categorized into cloze-type and prefix-type according to the position of the blank. Depending on the writing method, they can be classified into manual prompts written by a person and auto prompts which are generated automatically. We selected cloze-type prompts suitable for RoBERTa and manual prompts to perform our tasks with concise template sentences. We experimented with various prompts to generate user preferences and item attributes (key factors) from reviews. Owing to the nature of manual prompts, we observed that the key factor lists differed depending on the templates used in the prompts.

Table 1: Prompt used in the proposed method. [SEP] is a token used to separate sentences.

| | Prompt |
|------|--------|
| | [All reviews] [SEP] I value the [MASK]. |
| User | [All reviews] [SEP] I think the [MASK] is important. |
| | [All reviews] [SEP] I am a consumer who values the [MASK]. |
| | [All reviews] [SEP] The [MASK] of this item is important. |
| Item | [All reviews] [SEP] The point of this product is the [MASK]. |
| | [All reviews] [SEP] The [MASK] matters the most to customers. |

As our method does not have a ground truth (target label) for the generated key factor, we established criteria for template selection based on the performance of the key factor-based recommendation model and qualitative analysis, which confirmed the correlation between integrated reviews and generated key factors. Table 1 presents the template lists that we considered, and for each data category, we selected the most appropriate template from them.

Figure 4 illustrates examples of user and item key factor generation from reviews using our prompt and pretrained RoBERTa-base. The proposed method can generate user and item key factors that refine review texts. Additionally, we confirmed that these key factors allow us to identify user preferences and item attributes intuitively.

## 4.3 BASELINE METHODS AND EXPERIMENTAL SETTINGS

To validate the performance of the proposed method, we compared it with four baseline methods : **DeepCoNN**, **Transnet**, **D-attn**, **DAML**. The detail of baselines are presented in Appendix B.

We implemented the proposed methods and baselines using PyTorch. All models use 50-dimension word embeddings from Glove, and we followed the baseline model settings mentioned in their papers. We conducted a grid search to tune the values for the hyperparameters not explicitly specified in the studies. In our model, we use the top 10 key factors for both users and items. The hyperparameter for the latent feature dimensions is set to 50 for both the user and item networks, the dropout rate is 0.5, the optimizer used is Adam, the batch size is 128, and the initial learning rate is 0.002. As in prior research (Zheng et al., 2017), the performance metric used for the evaluation is the Mean Squared Error (MSE). All experiments were repeated five times, and we reported the average value of test MSE when the validation MSE was the lowest.

$$MSE = \frac{1}{n} \sum_{n=1}^{N} (r_{u,i} - \hat{r}_{u,i})^2 \tag{21}$$

---

[1]http://jmcauley.ucsd.edu/data/amazon

| | (example1) Automotive | | (example2) Office Products |
|---|---|---|---|
| User Reviews | (…) These are great, everyone should have them in the toolbox. I lost my old strap oil filter wrench and these are great for the replacement. Get one or two, you'll be happy you did. It is a battery in a box that will help you jump start vehicles. Love the plug and leave it charging. (…) I also like the fact it does not have all the gadgets the other unit out there have. They filled the space in this unit with a bigger battery! (…) It is an oil filter that is made by Motorcraft. I am now a believer of using Motorcraft parts whenever I can. They will not disappoint you as some other manufactures have me. Buy and you will be happy Neat little gadget. (…) It is a little large if you like checking tire depth from the top of the tire with a low profile vehicle. I will use it for many years as there are no moving parts. It may outlive me unless I lose it!! (…) Easy to use. I have not found any issues so far. (…) | User Reviews | This is what I discovered: 1) The tape dispenses easily and when I use it, noticed it stuck well to the package not to my fingers or hand like the cheap tapes. 2) The tape is thick enough to cover the area on the box I want it to cover and I don't have to keep on adding more tape to make it work. 3) (...) I rely on this superb tape to adjust out to the edges and cover them well. (...) Very pleased with this tape. Highly recommend it. Bought these very attractive and high quality folder to use for home office filing of important paperwork. (...) I use the Pilot Pen and the BLACK G2 Gel Ink refills for writings extensive reports for a club that I belong to. (…) I want them to be clear to read and as professional as possible. And these exceeded all of my expectations (…) These precise writing pen refills, I use for these reports as they have fine points which enhance my handwritten report greatly. (...) |
| User Key Factors (top-10) | price, design, warranty, simplicity, size, weight, cost, installation, quality, name | User Key Factors (top-10) | pen, precision, quality, detail, accuracy, ink, writing, speed, paper, pencil |
| Item Reviews | This is he first tire gauge (…) Surprisingly, all of the other ones that I've purchased were more expensive than this one, yet they all give inaccurate readings. In addition, the prior ones end up leaking out more air than I put in. (...) It's a pleasure to finally find such a simple gauge that actually works. I'm one of those that check tire pressure once a week so having a good tire gauge is imperative. (…) As soon as i got it i ran out and compared it to a mechanical one which I have been using with frustration and also a friend of mines gauge and low and behold this one read accurately to 35 lbs. (…) This gauge is simple to read, easy to get a seal w/o much pressure to the valve and comfortable to hold glove or not. (...) Easy to use. The light makes it much easier to read than other digital gauges. The pressure readings agree exactly with my other digital gauge. | Item Reviews | I really like this product. If you have to do a presentation, then this is the product for you. It keeps your information very organized, and easy to find. In my teaching of classes I use dividers to keep my lecture notes together and handy. (...) These are well made and nicely packaged. There are 5 sets in the box and they're kind of bundled together in packets which is good if you have people rifling through the supply closet looking for stuff. (...) Then the opportunity came along that offered these outstanding Wilson Jones Dividers! They are transparent and will never wear out in my lifetime! (...) These are "eight tab" dividers and five sets are nicely packaged in a sturdy package. (...) I file just about everything: receipts, bills (...) This boxed set is especially useful as it comes with 5 sets containing 8 tabs each. This makes it very convenient (...) |
| Item Key Factors (top-10) | light, simplicity, convenience, price, safety, ease, versatility, accuracy, value, warranty | Item Key Factors (top-10) | simplicity, presentation, packaging, price, convenience, documentation, versatility, transparency, storage, product |

Figure 4: Examples of user and item key factor generation. the left is a sample from the Automotive category, and the right is from the Office Product. The parts where reviews and key factors appear to be related are highlighted in bold and color.

## 4.4 PERFORMANCE EVALUATION

Table 2 summarizes the performance comparisons between the proposed method (Prompt2Rec) and the baselines for the five datasets. The proposed method exhibits a higher performance than the other baselines, which validates the effectiveness of our method. This is because baseline models (learning latent features using the entire review text) may not properly reflect user preferences and item attributes owing to noise in the review text. However, our method addresses this issue using the generated key factors as refined information.

Table 2: Performance comparisons between the proposed method and the baselines on five datasets.

| | Automotive | Amazon Instant Video | Office Products | Digital Music | Grocery & Gourmet Food |
|---|---|---|---|---|---|
| DeepCoNN | 0.8771 | 1.0902 | 0.7917 | 1.1015 | 1.1359 |
| Transnet | 0.7172 | 1.1407 | 0.7666 | 1.1731 | 1.1633 |
| D-attn | 0.7398 | 1.0880 | 0.7445 | 1.0708 | 1.1041 |
| DAML | 0.7678 | 0.9644 | 0.6419 | 0.8022 | 0.9493 |
| **Prompt2Rec** | **0.6951** | **0.9394** | **0.6308** | **0.7869** | **0.9437** |

## 5 MODEL ANALYSIS

We conduct an ablation study and a parameter study. (results of the parameter study are in Appendix C.) Furthermore, we visualize the attention weights to gain insight into the model's explainability.

Table 3: Performance comparison of the model variants

| | Automotive | Amazon Instant Video | Office Products | Digital Music | Grocery & Gourmet Food |
|---|---|---|---|---|---|
| w/o Attn | 0.7287 | 0.9735 | 0.6398 | 0.7975 | 0.9624 |
| only L-Attn | 0.7157 | 0.9609 | 0.6308 | 0.7892 | 0.9537 |
| only M-Attn | 0.7117 | 0.9445 | 0.6350 | **0.7859** | 0.9550 |
| Our Method | **0.6951** | **0.9394** | **0.6263** | 0.7869 | **0.9437** |

## 5.1 ABLATION STUDY

Table 3 shows the performance under four different setups: first, removing

both local and mutual attention (w/o Attn); second, using only local attention (only L-Attn); third, using only mutual attention (only M-Attn); fourth, our method using both local and mutual attention. Overall, our method exhibits the highest performance. This becomes a necessary condition for Explainability Analysis through the visualization of attention (Wiegreffe & Pinter, 2019).

## 5.2 EXPLAINABILITY ANALYSIS

To explain the model's recommendations, we visualize the attention weights. Our attention modules calculate local and mutual attention weights and then multiply them as in Equation (8); therefore, the final attention weight can be considered as the importance of words that reflect the similarity between user-item key factors. It is noted that, during the model training, the similarity (mutual attention) weights remain unchanged, and only the importance weights (local attention) are adjustable.

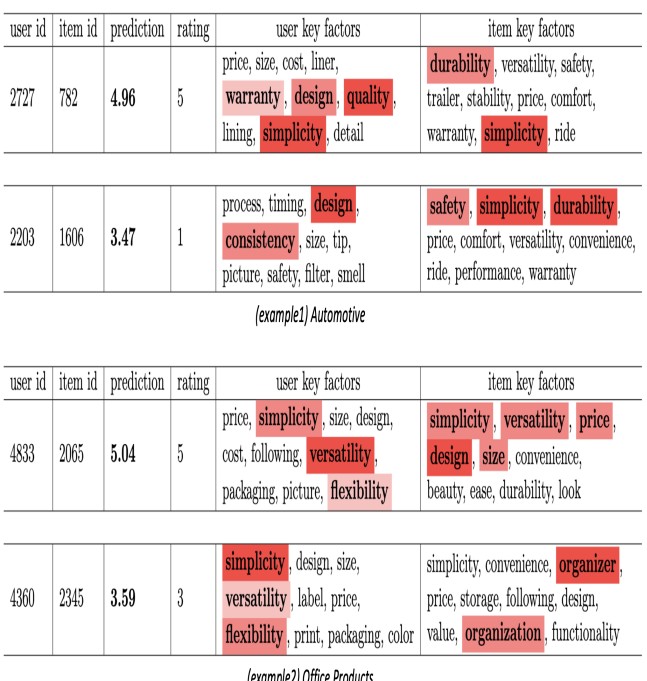

Figure 5: The top is a sample from the Automotive category, and the bottom is from the Office Product. The intensity of the color indicates a higher attention value on the key factor.

Figure 5 shows examples that highlight key factors with high attention weights when our model predicts ratings. To visualize this, we applied the softmax function to the attention weights and highlighted those higher than the median weight. At the top of Example 1, the model predicts a high rating, and we observe that the user key factors with high attention weights are (in descending order) [simplicity, quality, design, and warranty]. Regarding the item key factors, the model focuses on [simplicity and durability]. Semantically, there is a strong association between the key factors: (user key factor) simplicity matches with (item key factor) simplicity, and (user key factors) quality and warranty are similar with (item key factor) durability. while At the bottom of Example 1, the model predicts a low rating. We can observe that the association between the key factors with high attention weights is lower than that in the previous case. For Example 2, we can also infer the reasons for the prediction using the same approach as in Example 1.

Through explainability analysis, we confirmed that visualizing the attention weights on key factors can allows us to provide explanations for recommendations. Unlike previous research that visualizing attention weight on review texts to explain models (Zhang & Chen, 2020), our method, which utilizes refined key factors, provides more intuitive and user-friendly recommendation explanations.

## 6 CONCLUSION

We propose a novel user and item re-characterizing method called Prompt2Rec that introduces Prompt-based learning. It generates key factors that newly define essential user and item characteristics from review texts and uses them as new data sources to train the recommendation model. To validate the proposed method, we conducted quantitative and qualitative evaluations using five Amazon review 5-core datasets. The results show that our method achieves improvements over existing review-based deep learning recommendation models and has the potential to provide explanations for recommendations by visualizing the attention weights.

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

## A  DATASETS

Table 4: Datasets overview

| Dataset | Users | Items | Ratings |
|---|---|---|---|
| Automotive | 2,928 | 1,835 | 20,473 |
| Amazon Instant Video | 5,130 | 1,685 | 37,126 |
| Office Products | 4,905 | 2,420 | 53,258 |
| Digital Music | 5,541 | 3,568 | 64,706 |
| Grocery and Gourmet Food | 14,681 | 8,713 | 151,254 |

## B  BASELINES

- DeepCoNN: utilizes two parallel CNNs to extract latent features from user and item reviews. The features are combined and fed into a Factorization Machine (FM).
- Transnet: extends DeepCoNN by adding a teacher model to learn features for the target review. A student model is trained to be similar to the features of the teacher.
- D-attn: utilizes CNNs with global and Local attention to weight more important words from reviews. And the dot product between user-item features is performed.
- DAML: combines CNNs with local & mutual attention. Review-based and id-based features are combined and passed into the Neural Factorization Machine (NFM).

Figure 6 illustrates the differences between the proposed method (Prompt2Rec) and baselines. Our method trains the recommendation model using key factors generated through prompt-based learning using RoBERTa-base. In contrast, the baseline models are trained using the review text itself.

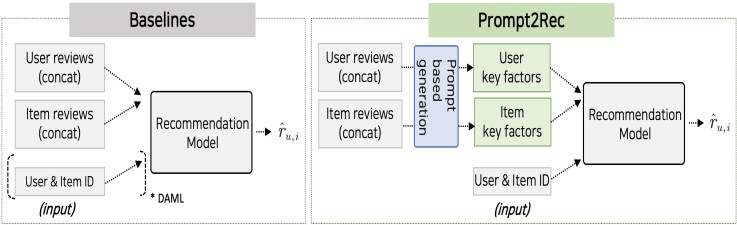

Figure 6: The differences between methods

## C  PARAMETER ANALYSIS

### C.0.1  NUMBER OF KEY FACTORS (TOP - K)

Table 5 shows the performance of the model when the number of user and item key factors (top-k) was changed to [5, 10, 20]. The model performance tended to be higher when 10 key factors (top-10). We believe that too few or too many data may negatively impact the learning of user and item features. Hence, we chose to use ten key factors for the performance and efficiency of the explainability analysis.

Table 5: Effect of the number of key factors (Top - k)

| | Automotive | Amazon Instant Video | Office Products | Digital Music | Grocery & Gourmet Food |
|---|---|---|---|---|---|
| Top-5 | **0.6720** | 0.9624 | 0.6320 | 0.7922 | 0.9526 |
| Top-10 | 0.6951 | **0.9394** | 0.6308 | **0.7869** | **0.9437** |
| Top-20 | 0.7121 | 0.9432 | **0.6263** | 0.7911 | 0.9493 |

### C.0.2  NUMBER OF LATENT FACTOR DIMENSION

Similar to other studies, we examined the effects of the latent factor dimensions. Table 6 presents the performance when the latent factor dimension was changed to [10, 25, 50, 100]. We did not observe any performance improvement with an increase in the latent factor dimensions, and the model showed the highest performance when the latent factor dimensions were 25 or 50. This suggests that increasing the number of learning parameters might enhance the model's representational capacity; however, it could also lead to overfitting of the training data, resulting in a decrease in the generalization performance of the test data. Therefore, we set the latent factor dimension to 50.

Table 6: Effect of the number of latent factor dimension

|  | Automotive | Amazon Instant Video | Office Products | Digital Music | Grocery & Gourmet Food |
|---|---|---|---|---|---|
| Dim-10 | 0.6904 | 0.9500 | 0.6316 | 0.8086 | 0.9612 |
| Dim-25 | **0.6834** | 0.9452 | 0.6348 | 0.7955 | **0.9394** |
| Dim-50 | 0.6951 | **0.9394** | **0.6263** | **0.7869** | 0.9437 |
| Dim-100 | 0.6996 | 0.9422 | 0.6319 | 0.7985 | 0.9564 |

