# OpenReview forum: "Prompt2Rec : Prompt based user and item Re-characterizing method for Recommendation"
_ICLR.cc/2024/Conference — Submitted to ICLR 2024_

### Official Review · Reviewer_EWwu · 2023-10-30

**Soundness:** 2 fair
**Presentation:** 3 good
**Contribution:** 1 poor
**Rating:** 3
**Confidence:** 5

**Summary:**

The paper proposes leveraging LLMs, specifically RoBERTa, to extract essential information from textual review without fine-tuning. For each review, the authors use several cloze-test style prompts to extract various information, such as “[All reviews][SEP] I value the [MASK].”. The information is then used to train a CNN-based recommendation system, which consists of local a word attention module, a CNN module, user-item feature fusion module, and a neural factorization machine for rating prediction. They study the attention weights of the local word attention module, showing that the attention weights can be focused on representative words.

**Strengths:**

1. The paper is well-written and easy to follow.
2. Large-language models has shown impressive performance on understanding textual data. Exploring LLMs is a promising direction in recommender systems.

**Weaknesses:**

1. The core idea of the paper is using LLMs for extracting essential information from reviews such as its keywords. However, it has limited novelty due to the extensive prior exploration of using LLMs for information extraction. Suggest referencing the survey paper. [1]
2. Roberta is not a very large language model. It makes more sense to me to directly fine-tune the Roberta rather than only using it to extract the key information without fine-tuning. With supervised training signal, I believe fine-tuning Roberta can better filter out noisy information in reviews. This should be one of the baselines. In addition, directly using LLMs as a recommender system has been explored in previous work, and this line of work should be discussed and compared. [2][3][4]
3. The comparison of using different keyword or keyphrase extraction methods such as TF-IDF should be included.

References:
[1]A Survey on Recent Advances in Keyphrase Extraction from Pre-trained Language Models, Song et. al.
[2]Language Models as Recommender Systems: Evaluations and Limitations, Zhang et. al.
[3]Zero-Shot Next-Item Recommendation using Large Pretrained Language Models, Wang et. al.
[4]M6-Rec: Generative Pretrained Language Models are Open-Ended Recommender Systems, Cui et. al.

**Questions:**

1. Instruction-tuned language models such as Llama2 have shown better performance on prompt-based learning than Roberta. Why these models were not chosen?
2. How does using different prompts affect the model performance? Are there large differences in performance using different prompts?

---

### Official Review · Reviewer_TbAk · 2023-11-01

**Soundness:** 2 fair
**Presentation:** 1 poor
**Contribution:** 1 poor
**Rating:** 1
**Confidence:** 5

**Summary:**

The paper presents a method to extract user feature and item features from reviews based on RoBERTa, where the features are some word factors that a user may care about, and then feed such feature word embeddings into a neural network model (CNN) for recommendation score calculation. Experiments are conducted on several e-commerce datasets and compared with some text-based recommendation baselines.

**Strengths:**

The paper tries to explore prompt-based recommendation which is a trending topic, though the paper does not mention any other existing prompt-based recommendation models.

**Weaknesses:**

The motivation is somewhat weak: The key motivation of the research is to remove "noise" from user review data, i.e., if all review texts are used, there is likely to be a significant amount of irrelevant or unnecessary data (noise) unrelated to user preferences or item attributes. However, the attention mechanism of modern deep learning architectures such as Transformers and LLMs are exactly developed to solve the problem, by automatically learning the importance of difference parts of a text so that we don't need to manually select the parts from text. The proposed method actually goes back to devoting efforts to purposely selecting some parts of text for model learning, which is exactly what modern AI models tries to avoid.

There have been many prompt-based methods for recommendation, however, the paper did not compare with or even mention any of them.

The related work section is messed up: (1) It cites a paper published in 2022 as an "early study" of using topic models for review processing, and then cites a paper published in 2017 to criticize the weakness of the 2022 paper. (2) There have been many prompt-based models for recommender systems, but the related work does not mention any of them. The experimentation does not compare with any existing prompt-based recommendation model either.

The proposed model does not provide much insight for the community, it uses some prompts to extract some features from reviews, and then feed these feature embeddings into a traditional recommendation model. It's not like a research paper but more like a class project report.

The experiments used MSE as the evaluation metric, however, it has been widely known that MSE (or RMSE) cannot reflect the real ranking performance of recommender systems. It is important to evaluate the proposed method based on widely recognized ranking metrics such as Hit Ratio, NDCG, precision, recall, F1, etc.

**Questions:**

What is "re-characterize", what are "key factors"? what factors are key and what factors are not key?

In this sentence: "Prompts are designed to enable the generation of key factors that newly define user preferences and item attributes from review text.", authors used the word "newly define"; in the following sentence: "The key factors generated by Prompt2Rec are essential words representing user preferences (or item attributes) that newly refine an integrated review text.", authors used "newly refine". Are "newly define" and "newly refine" talking about the same thing or different things?

---

### Official Review · Reviewer_EFBG · 2023-11-05

**Soundness:** 3 good
**Presentation:** 2 fair
**Contribution:** 3 good
**Rating:** 6
**Confidence:** 4

**Summary:**

The authors propose an LLM prompting-based approach for collaborative filtering. The main idea is to learn key factors to summarize the main characteristics of users and items from review texts, and then use this new information to train a model for predicting review ratings.

**Strengths:**

Using the extracted key factors as part of input were shown to improve the performance on the Amazon product review datasets. Augmenting recommendation with key factors extracted from review data is an interesting idea.

**Weaknesses:**

One question about the proposed approach is its generalizability. Why is it necessary to divide the Amazon review datatset into 5 five datasets for different product types? Can we combine the datasets together so that the method become more generally applicable?

The key factors are embedded by Glove embeddings. Given there are more advanced transformer-based embeddings in recent years. Have the authors tried these advanced embeddings? Why just Glove embeddings?

The application in the paper is review ranking prediction. It is a bit misleading to claim that it is a recommendation model, where behavior data such as click, purchase, add-cart actions are more of the focal point, and items are recommended for users to consume. I don’t think claiming recommendation is essential for the paper.

**Questions:**

One question about the proposed approach is its generalizability. Why is it necessary to divide the Amazon review datatset into 5 five datasets for different product types? Can we combine the datasets together so that the method become more generally applicable?

The key factors are embedded by Glove embeddings. Given there are more advanced transformer-based embeddings in recent years. Have the authors tried these advanced embeddings? Why just Glove embeddings?

---

### Official Review · Reviewer_dWh3 · 2023-11-06

**Soundness:** 2 fair
**Presentation:** 2 fair
**Contribution:** 2 fair
**Rating:** 3
**Confidence:** 4

**Summary:**

The author addresses the issue of sparse interaction data in collaborative filtering by leveraging advanced NLP techniques to extract critical user preference indicators and item attributes from review texts. Rather than using entire reviews, key factors are distilled using these texts as prompts for a pre-trained model, mitigating the issue of noisy data. Additionally, the author enhances the model's transparency by visualizing its attention weights on these key factors, thereby offering explanations for its recommendations.

**Strengths:**

- The author builds upon prior research that identifies user and item IDs as crucial for uncovering latent information, innovatively enriching this information with insights from review texts.
- The novel approach of using review texts as prompts to generate key factors for a pre-trained model enhances the recommendation system’s relevance and accuracy.
- Visualization of attention weights provides an explanatory layer to the recommendation process, offering valuable insights into model decisions.

**Weaknesses:**

- The paper could be strengthened by including an ablation study that isolates the impact of using the Neural Factorization Machine model without review text integration.
- Readers are left without a clear benchmark of improvement attributed to the proposed method as there is no comparative analysis between the baseline model and the enhanced model with review information.
- There is an assumption that key factors generated from reviews will always enhance recommendation quality, which may not account for potential biases or inaccuracies in the review content itself.
- No top-k evaluations are conducted, which should be more common in RS.

**Questions:**

1. Why did the author opt for GloVe embeddings over RoBERTa for generating key factor embeddings? Are there specific concerns or desired GloVe characteristics at play?
2. Would the model's performance improve by limiting the number of key factor tokens to reduce noise and increase generation efficiency?
3. What criteria are used for selecting words within the key factor corpus?
4. ow does the variety of key factors influence the model's performance?

---

### Official Review · Reviewer_ZfrZ · 2023-11-10

**Soundness:** 3 good
**Presentation:** 3 good
**Contribution:** 2 fair
**Rating:** 3
**Confidence:** 3

**Summary:**

This paper introduces the prompt-based learning paradigm of Natural Language Process (NLP) for re-characterizing users and items in rating prediction problem in recommendation. The paper proposes to use a pre-trained language model to generate user and item characteristics (also called key factors in the paper) from review text and use such key factors for training a rating prediction model. The paper empirically demonstrates that using key factors improves the performance of existing rating prediction models using review text. The paper also argues that the attention weights associated with key factors can be used as explanations for recommended items.

**Strengths:**

1. The related work is well-done.
2. The proposed method (termed Prompt2Rec) beats all baseline models in terms of Mean Squared Error (MSE) in five widely used benchmark datasets.
3. The paper provides a few good case studies showing that the key factors generated by the pretrained language model (RoBERTa) are intuitive and explainable.

**Weaknesses:**

1. My major concern is the novelty of the paper. The pretrained lagnuage model RoBERTa is simply used as a black box and the template used to construct prompt looks straightforward. The ways how the proposed method Prompt2Rec uses the generated key factor as inputs and trains a rating prediction model are also very standard.
2. Presentation of the paper can be improved. E.g., the words in Figure 1 and Figure 4 are very small and it is pretty difficult to read.

**Questions:**

1. How many templates do you use for constructing prompt and how do you obtain the templates?

---

### Meta-Review · Area_Chair_FbmN · 2023-12-06

**Metareview:**

I recommend to reject this paper for ICLR.

In this paper, the authors propose Prompt2Rec, which leverages a prompt based pre-trained Roberta on review texts to generate key factors for both users and items and trains a rating prediction model using neural factorization machines. Experiments are conducted on the Amazon Review dataset.

All the reviewers recognize that the paper is well-written and easy to follow. However, the majority of them also think that   the novelty of this proposed method is limited for ICLR and the experimental settings should be improved, such as recommendation methods using more recent LLM models and the comparison of using different keyword or keyphrase extraction methods. I would suggest the authors to take the feedbacks from reviewers to further enhance the paper.

**Justification For Why Not Higher Score:**

I have read the paper and shared with the same evaluation with the majority of reviewers that this is a clear reject paper.

**Justification For Why Not Lower Score:**

N/A

---

### Decision · Program_Chairs · 2024-01-16

Reject